# Human Microglia Models for NeuroHIV

**DOI:** 10.3390/v17050641

**Published:** 2025-04-29

**Authors:** Priyanka Sarkar, Xu Wang, Wenhui Hu, Jian Zhu, Wen-Zhe Ho

**Affiliations:** 1Department of Pathology and Laboratory Medicine, Temple University Lewis Katz School of Medicine, Philadelphia, PA 19140, USA; priyanka.sarkar@temple.edu (P.S.); xuwang@temple.edu (X.W.); 2Department of Neuroscience, Virginia Commonwealth University, Richmond, VA 23298, USA; wenhui.hu@vcuhealth.org; 3Department of Pathology, Ohio State University Wexner Medical Center, Columbus, OH 43210, USA; jian.zhu@osumc.edu

**Keywords:** HIV, microglia, peripheral blood monocyte-derived microglia (MMG), induced pluripotent stem cell (iPSC)-derived microglia (iMg), microglia-containing cerebral organoids (MCOs) from iPSCs

## Abstract

Microglia are the primary target and reservoir of HIV infection in the central nervous system (CNS), which contributes to HIV-associated neurocognitive disorder (HAND). However, studying HIV infection of microglia has been challenged by the limited availability of primary human microglial cells. To overcome this issue, investigators have developed various microglial models for HIV studies, including immortalized human microglial cell lines, HIV latently infected microglial clones, peripheral blood monocyte-derived microglia (MMG), induced pluripotent stem cell (iPSC)-derived microglia (iMg), and microglia-containing cerebral organoids (MCOs) from iPSCs. Though these models have been used in many laboratories, the published data about their expression of the specific human microglia markers and the HIV entry receptors are conflicting. In addition, there is limited information about their feasibility and applicability as a suitable model for acute and/or latent HIV infection. This review provides a concise summary of the currently used human microglial models, with a focus on their suitability for NeuroHIV research.

## 1. Introduction

As the primary immune cells in the brain, microglia play a crucial role in the brain’s immunity against viral infections, including HIV [1]. However, because microglia express the HIV entry receptors (CD4, CCR5, and CXCR4), they are also the major target for both productive and latent HIV infection in the brain [2,3,4]. As long-lived resident cells in the brain, microglia serve as a reservoir and allow persistent and latent HIV infection. Infected microglia produce HIV proteins and inflammatory cytokines, resulting in neuroinflammation and HIV-associated neurocognitive disorders (HANDs) [5,6]. Although combination antiretroviral therapy (cART) is effective in suppressing HIV replication and decreases the 40–50% of dementia cases [7], many patients continue to have cognitive impairments due to persistent HIV infection in the brain [4,5,8]. Therefore, it is essential to understand the role of microglia in the immunopathogenesis of NeuroHIV infection. However, studies on primary human microglia have been hampered by the difficulties of obtaining surgical human brain specimens, isolating a high quality/quantity of microglia, and the cells’ short lifespan in culture. To overcome these issues, researchers have developed various human microglial culture models for NeuroHIV studies (Table 1). This review focuses on these models that have been commonly used by different laboratories, including primary human microglia, microglial cell lines, peripheral blood monocyte-derived microglia (MMG) or MMG-like cells, HIV latently infected microglial cell lines, human induced pluripotent stem cell (iPSC)-derived microglia (iMg), and microglia-containing cerebral organoids (MCOs) from iPSCs.

**Table 1 viruses-17-00641-t001:** An Overview of Microglial Model Development.

	1995	2001	2017	2016	2017	2016–2018	2013	2020
Model	Human microglia line		Monocyte-derived microglial cells	Microglia lines for latent infection		iPSC-derived microglia	Cerebral organoids	Monocyte-containing cerebral organoids from iPSC
	HMC3 [6,9,10]	HMO6 [11,12,13]	MMG [14]	C20 [15]	HC69.5 [16,17,18]	iMg [6,10]	Cos [19]	MCOs [6,20,21]
Microglia markers	IBA-1, P2RY12	CD11b, CD68, CD86, HLA-DR, HLA-ABC	CD11b, CD11c, CD80, IBA-1, P2RY12	IBA-1 CD64 CD86,	P2RY12, CD11b	P2RY12, TREM2, IBA-1, CD11b, TMEM119	NA	AIF1, TMEM119, TREM2, P2RY12
HIV entry	CD4-	CD4-	CD4+	CD4-	CD4-	CD4+	NA	CD4+
receptors	CCR5+	NA	CCR5+	CCR5+	CCR5-	CCR5+	NA	CCR5+

N/A: data not available.

## 2. Primary Human Microglia

Primary human microglia can be isolated from the brain tissue of a deceased person and cultured for several weeks under microglia-specific enrichment conditions [6,22]. Other sources of primary human microglia are fetal brain tissues following abortions and patients undergoing surgery for brain tumors or epilepsy [23,24,25]. The technique protocols for such isolation have been well-established [22,26,27,28]. Like primary human macrophages derived from peripheral blood monocytes, in vitro-cultured microglia are highly susceptible to HIV infection [29], and infected cells produce inflammatory cytokines [2,30]. However, obtaining fresh human brain tissue to isolate large quantities of viable microglia is extremely challenging. Additionally, the heterogeneity of samples, obtained from different brain regions of different donors (differing in age, sex, health condition, etc.), can result in many confounding variables for experiments with primary human microglia. Therefore, it is crucial to develop alternative in vitro human microglial models for HIV studies.

## 3. Microglial Cell Lines (HMC3 and HMO6)

HMC3 resembles the adhering characteristics of primary human microglia but has low phagocytic activity [9]. The cells express some of the microglial markers and antigenic molecules, such as IBA-1, CXCR1, CXCR3, NSE, and MHC-II/IFN-ɣ. However, they do not express CD14, CD68/Ki-M6, CD11c, and P2RY12 [10,31,32]. Importantly, HMC3 does not express the major HIV entry receptor CD4 and is not permissive for productive HIV infection/replication. Studies on HIV infection with HMC3 are limited to experiments with the pseudo-typed viruses [10,31]. Moreover, it has been reported that these cells could be of rat origin [15,33]. Nagai et al. developed another immortalized microglial cell line, named HMO6. Although HMO6 expresses the microglial markers (CD11b, CD68, CD86, HLA-ABC, HLA-DR, and RCA-1 lectin), these cells have a less diverse profile of secreted soluble inflammatory mediators than primary human microglia [11]. Like HMC3 cells, HMO6 lacks the expression of microglial markers (CD14 and CD11c) and CD4 receptors [12], and thus, their use for HIV studies is limited [6,11,12,13].

## 4. Microglial Cell Line for Latent HIV Infection (C20 and HC69.5)

Garcia-Mesa et al. [15] developed an immortalized microglial cell line (C20) for HIV studies. C20 was generated from primary glia isolated from human adult brain tissues and frozen glial cells. The cells have a microglia-like morphology and express the key microglial markers (CD11b, TGFβR, and P2RY12). Importantly, their RNA expression profiles have similar characteristics to primary human microglial cells. C20 expresses CCR5 but has an extremely low level of CD4 receptor expression, which decreases with increasing cell passages. Therefore, the cells are not suitable for productive HIV infection [15]. However, David Alvarez-Carbonell et al. [16] used C20 as the parental cell line to establish an HIV latently infected cell line (HC69.5). Briefly, HC69.5 was developed by immortalizing the C20 cell line with simian virus 40 large T antigen/human telomerase reverse transcriptase [16]. These immortalized cells were then transfected with a vesicular stomatitis virus G envelope pseudo-typed lentiviral vector with a green fluorescent protein as a reporter [16]. HC69.5 expresses the specific microglial markers (P2RY12 and CD11b) and the macrophage lineage marker (CD14). Importantly, HC69.5 cells express the HIV RNAs and viral proteins (Tat, Rev, Env, Vpu, and Nef) but lack HIV gag [15,34]. HC69.5 has been widely used as a latently infected HIV microglial model, which could be significantly activated by TNF-α [17,18]. Studies show that these cells express low levels of HIV (1–6%), but TNF-α induction for 16h could significantly induce HIV replication in 90% of the cell population [16]. Additionally, autocrine expression of TNF-α can spontaneously reactivate HIV in HC69.5 cells, which can be blocked by treatment with the glucocorticoid receptor agonist dexamethasone [17,18].

## 5. Human Peripheral Blood Monocyte-Derived Microglia (MMG)

Rawat et al. developed a protocol for generating peripheral blood monocyte-derived microglia (MMG) [14], which was based on the early work of MMG-like cells by Leone et al. [35]. They demonstrated that human peripheral blood monocytes cultured in a serum-free medium with M-CSF, GM-CSF, NGF, and CCL2 could differentiate into MMG. These cells are different in morphology, phenotype, and function from freshly isolated monocytes. They resemble primary human microglia and express microglial markers, including CD11b, CD11c, CD14, CD45, CD195, CD80, P2RY12, IBA-1, CD14, and CD45. MMG also express low levels of HLA-DR and CD86. Importantly, MMG express the key HIV entry receptors (CD4, CCR5, and CXCR4) [10] and could be productively infected with HIV and support long-term infection while continuously releasing virions into the culture media [10,14].

The reported protocols for generating MMG are relatively simple, feasible, and reliable. Ohgidani et al. showed that the addition of microglial-growth factors (GM-CSF and IL-34) to human monocyte cultures for 2 weeks can convert monocytes into microglia-like cells, which express CD11b (high)/CD45 (low) and CX3CR1 (high)/CCR2 (low) [36]. These cells release pro-inflammatory/anti-inflammatory cytokines and can perform phagocytosis [36]. Subsequent studies on MMG showed that additional factors, such as M-CSF, nerve growth factor (NGF)-β, and CC chemokine ligand 2 (CCL2), were suitable for HIV infection [14,37]. Sheridan et al. adapted this protocol for differentiating microglia from umbilical cord blood-derived mononuclear cells [38]. While MMG have been widely used as a microglial model for HIV studies, there is a concern about whether the added growth factors and cytokines could interfere with the study outcomes because some of these factors are known to be implicated in HIV infection.

## 6. Human Induced Pluripotent Stem Cell (iPSC)-Derived Microglia (iMg)

The discovery of induced pluripotent stem cells (iPSCs) has provided a platform for the generation of a wide variety of brain cell types, including human microglial cells (iMg) [39,40,41,42]. iMg are like primary human microglia in their morphology, gene expression, and cytokine release profile. They are distinct from other tissue macrophages as they display a profile of neuronal-co-culture-specific expression and inflammatory response. The iMg model has been utilized to study neurological diseases [43], such as Alzheimer’s disease [44], Parkinson’s disease [45], amyotrophic lateral sclerosis, and frontotemporal dementia [46]. Additionally, iMg can be infected by the Zika and dengue viruses [47]. Importantly, iMg expresses the microglial markers (P2RY12 and TMEM119) and the HIV entry receptors (CD4, CCR5, and CXCR4) [10]. Several groups have reported that iMg could be productively infected with HIV, particularly with CCR5-tropic strains [6,10,48]. We recently reported that iMg possess immunologically functional toll-like receptor 3 (TLR3), which could be activated by Ploy (I:C), and produce the antiviral cellular factors against HIV [49]. Although iMg have been successfully used in HIV infection studies, there is little information on establishing persistent/latent HIV infection in these cells, which is likely due to their short in vitro lifespan. The protocols for generating human iMg have been well established [50]. Mcquade et al. published a simplified protocol for establishing iMg [51]. Additionally, several companies have now provided detailed protocols; the culture media and iMg cells that originate from different donors (Table 2).

## 7. Microglia-Containing Cerebral Organoids (MCOs) Derived from Human iPSCs

In the in vivo microenvironment, microglial functions significantly depend on their direct and/or indirect contact with other brain cell types, such as neurons and astrocytes. Therefore, it is clinically important to develop a microglia-containing cellular model with other key brain cells. In 2013, Lancaster et al. [19] reported the development of cerebral organoids (COs) from iPSCs. Since then, the field of iPSC-derived COs has been significantly advanced [52,53,54,55,56]. The major advantage of COs is that the cultured cells can self-organize into 3D structures and differentiate into the key major brain cell types, which recapitulate the layered structure, cellular diversity, and synaptic connectivity of the human brain [57,58]. Recently, human iPSC-derived COs have been increasingly used as a brain model for studying various neurological disorders and neurotropic virus infections. Importantly, it has been documented that iPSC-derived COs can be cultured for many months, and the longest duration of maintaining COs in culture was 800 days [59]. This feature of COs allows long-term studies on neurodevelopment or disease progression, which is particularly important for studying latent HIV infection.

Although iPSC-derived COs recapitulate some key features of human brain development, many of the currently used COs are derived from neuroectodermal progenitors and only contain neurons and astrocytes. They do not have microglia, which arise from mesodermal progenitors. Therefore, the absence of microglia in COs substantially limits their value and applicability for brain research, particularly HIV studies. As resident immune cells in the brain, microglia are crucial not only for brain immunity but also for neurogenesis and neuroinflammation. More importantly, microglia are the primary target and reservoir for HIV infection. Park et al. developed microglia-sufficient brain organoids by co-culturing COs with primitive-like macrophages generated from human iPSCs [60]. They demonstrated that iPSC-derived microglia promote organoid maturation via cholesterol transfer [60]. We and others have developed a protocol to generate microglia-containing cerebral organoids (MCOs) [54,61,62,63], demonstrating that MCOs express microglial markers (P2RY12 and TMEM112) and the major HIV entry receptors (CD4 and CCR5).

HIV infection of the MCOs model was first reported by Dos reis et al. [64]. They incorporated HIV-infected primary human microglia or the microglial cell line (HMC3) into COs. They demonstrated that this model supported low levels of HIV replication and that the HIV-infected microglia can produce inflammatory factors in COs [64]. Another group, Gumbs et al. [6], demonstrated that both MCOs and isolated organoid-derived microglia could be productively infected with replication-competent HIV-Bal reporter viruses. They found that the susceptibility of organoids to HIV infection was associated with the expression of the microglial marker (AIF1) and the HIV entry receptors (CD4 and CCR5), regardless of organoid maturation. Other groups also reported that productive HIV infection was only observed in microglial cells, which was dependent on the co-expression of microglia-specific markers and the CD4/CCR5 receptors [10,20,21]. More recently, Donadoni et al. showed that HIV replication in MCOs could be inhibited by cART [20]. We also observed that the MCOs from some human iPSC lines could be acutely infected by the live HIV Bal strain (unpublished data). In addition, in agreement with the study by Gumbs [6], we found significant variability between organoids from the same batch and across iPSC lines in terms of susceptibility to HIV infection. Furthermore, there are no data showing that MCOs support persistent/latent HIV infection (Table 3). These issues highlight an important limitation of the brain organoid model for HIV infection studies.

**Table 3 viruses-17-00641-t003:** HIV Infection of iPSC-Derived Brain Organoids.

Model	iPSC Origin	Organoid Age	HIV Strain	HIV Infection		Microglia Markers *	HIV Receptor	Reference
				Acute	Latent			
MG-hBORG, hBORG	fetal brain-derived neural progenitor cells	Day 30	NL (YU2-Env)- EGFP strain	peaked at NA day 11 post- infection	NA	IBA1	NA	[64]
o-MG	fibroblast	Week 1	Bal	peaked at day 6 post- infection		IBA1, AIF1, TMEM119, P2RY12, TREM2, CSF1R, CX3CR1	CCR5+, CD4+, CXCR4+	[20]
CEREBRAL AND CHOROID PLEXUS [ChP] BRAIN ORGANOID	mixed culture of wild-type iPSCs and modified iPSCs programmed for microglia differentiation	Day 14	ADA	peaked at day 30 post- infection	NA	IBA1 TREM2	CCR5+, CD4+, CXCR4+	[21]
CO-iMs	Hematopoietic progenitor and fibroblast	Day 50	Bal and Gag- iGFP-JRFL	peaked at day 5 post- infection	NA	TMEM119, IBA1, CX3CR1, CSF1R, P2RY12	NA	[65]
HUMAN NEUROSPHERES	neural Progenitor Cells (NPCs)	Week 12–14	89.6, JRCSF and CH040	peaked at day 14 post- infection	NA	IBA1	NA	[66]

* Microglial markers: mRNA and/or protein expression. NA: information not available.

## 8. Discussion: Pros and Cons of Human Microglial Models

This review compares the human microglial models for HIV studies. In Table 4, we summarize the advantages (pros) and disadvantages (cons) of these models regarding their feasibility and applicability to acute and persistent HIV infection. While the immortalized cell lines (HMO6 and HMC3) are readily available and resemble some aspects of the primary human microglia, their application, particularly for HIV infection, is limited because they do not express the major HIV entry receptor, CD4 (Table 1 and Table 4) [6,9,10,30,31,32,33]. Among the immortalized cell lines, only HC69.5 has been used as a human microglial model for studying HIV transcriptional latency and reactivation [17,34].

In addition to the microglial cell lines, human peripheral blood monocyte-derived microglia (MMG) have been used as a microglial model for HIV infection. However, there is concern about whether the addition of the microglial growth factors and cytokines to the cultures can introduce variables that affect the study outcomes. Some of these factors are known to be implicated in regulating HIV infection/replication and the innate immune function of these cells. Recently, the iPSC-derived microglia (iMg) model has gained great attention from investigators because, like primary human microglia, iMg can be productively infected with HIV [6,10,49]. In contrast with immortalized microglial cell lines, primary models like MMG and iMg more closely resemble human microglia, with transcriptional, immunological, and morphological features that mimic primary human microglia. More importantly, MMG and iMg express the HIV entry receptors (CD4, CXCR4, and CCR5) and are highly susceptible to HIV infection. However, MMG and iMg are fully differentiated cells with a limited lifespan, which limits their use for the study of persistent and latent HIV infection. In addition, MMG, in particular, seems to be highly cytokine-dependent [6,9,17,18,47].

Finally, in comparison with 2D cultures of the microglia models, 3D cerebral organoid models, particularly MCOs, are more advanced in vitro models for studying HIV brain infection and infection-mediated neuropathogenesis. These models provide a brain-like environment; however, they lack vasculature and blood–brain barrier functions. Yet, various groups (Table 3) have shown that MCOs could be productively infected by HIV and produce inflammatory cytokines, specifically in the presence of other brain cell types [20,21,50,64,65,66,67]. However, while MCOs have a significantly longer in vitro lifespan than iMg, there is little information about whether MCOs support persistent/latent HIV infection (Table 4). In addition, both COs and MCOs do not contain the blood–brain barrier (BBB) and perivascular macrophages, another major target of HIV in the brain. Moreover, the shape, size, and maturing time point of MCOs vary between iPSC line donors and culture batches. During long-term cultures, the core of COs or MCOs cannot obtain sufficient nutrients/oxygen, resulting in cell death within organoids. Therefore, overcoming these limitations [56] is essential for further improving the brain organoid models for NeuroHIV studies.

In summary, although the currently used human microglial models do not fully recapitulate the scenarios in the brain micro-environment, a combination of these models together would allow researchers to target distinct stages of the HIV lifecycle, ranging from entry/replication (iMg and MMG) to latency/reactivation (HC69.5) and neuroinflammatory signaling and multiple brain cell interactions (MCOs). In general, by leveraging the complementary strengths of different microglial models, researchers can gain a more comprehensive understanding of the complex interplays between HIV, microglia, and other brain cells. However, it is necessary to further improve and develop human microglial models for NeuroHIV studies.

**Table 4 viruses-17-00641-t004:** Microglial Models for HIV.

Model	Microglia Markers	CD4	CCR5	CXCR4	HIV Infection		Pros	Cons	References
					Acute	Latent			
Primary Human Microglia	+	+	+	+	+	+/−	Acute HIV infection	Limited availability	[5,6,17]
Microglia Lines (HMC3, HMO6)	+	−	+	+	−	−	Microglia function	No CD4	[10,11,12,13,15,31,32,33]
HIV Latently Infected Microglia Line (HC69.5)	+	+/−	−	+	−	+	HIV latency activation	No CD4	[16,17,18]
Peripheral Blood Monocyte- derived Microglia (MMG)	+	+	+	+	+	+/−	Acute HIV infection	Added growth factors might affect HIV infection	[14,35,36]
iPSC-derived Microglia (iMg)	+	+	+	+	+	+/−	Acute HIV infection	Donor variability, limited quantity	[6,20,44,45]

+/−: it is unclear.

## Figures and Tables

**Table 2 viruses-17-00641-t002:** Commercial Availability of human iPSC-Derived Microglia.

Company	Format	Culture Media/Protocol Availability	iPSC Origin	Microglia Markers
**Applied Stem cell (Milpitas, CA, USA)**	cryopreserved, fully differentiated	Yes	fibroblasts from Caucasian/African American male	P2RY12, CX3CR1, TMEM119, and IBA1
**Axol Biosciences (Cambridge, UK)**	cryopreserved, mature microglia	Yes	monocytes from 40–50 years old male donor	TREM2, IBA1, and TMEM119
**Fujifilm Cellular Dynamics Inc. (Madison, WI, USA)**	frozen, differentiated	Yes	fibroblasts and PBMC from a female/male Caucasian donor	TREM2, and IBA1
**Bit.Bio (Cambridge, UK)**	cryopreserved, immature	No	skin fibroblast from Caucasian adult male and female	TMEM119, IBA1, CD11b, CD45, P2RY12, TREM2, CX3CR1

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
