# Peer review of "Human Microglia Models for NeuroHIV"

_viruses, 2025, doi:10.3390/v17050641_

Round 1
Reviewer 1 Report
Comments and Suggestions for Authors
The review article by Sankar and colleagues nicely summarizes the microglia models available to study HIV replication in the CNS. This manuscript is timely and would be of interest for researchers working on HIV and other neurotropic viruses. However, this work will largely benefit from an improved discussion section including additional pros and cons related to the use of immortalized cell lines versus primary and/or primary-like cultures. Indeed, while we have a vast knowledge on HIV replication in T-cells, there are several gaps in our understanding of HIV replication in human microglia. This last comment not only applies for the establishment of latency but also for the different steps of active replication. Authors should also comment on this explaining how the available models could be exploited to improve our understanding on the HIV - microglia (and CNS) interaction.
Once revised, the manuscript will be suitable for publication in Viruses
Author Response
We appreciate the reviewer’s positive comment, “This manuscript is timely and would be of interest for researchers working on HIV and other neurotropic viruses”.
Major Comments
Comment 1: This work will largely benefit from an improved discussion section, including additional pros and cons related to the use of immortalized cell lines versus primary and/or primary-like cultures.
Response: Thank you for the comments and suggestions. We have now included additional pros and cons regarding the microglia lines versus primary and/or primary-like cultures in the manuscript (Lines 203-210, and 218-225).
Comment 2: This last comment not only applies for the establishment of latency but also for the different steps of active replication.
Response: The various microglia models described in the paper can be used for studying different stages of the HIV replication cycle, including the establishment of latency. For acute HIV infection, both primary human microglia and primary-like cultures such as MMG and iMg are suitable to study the full HIV replication cycle, from viral entry to productive infection and release of new virions. These models express the key HIV entry receptors (CD4, CCR5, CXCR4) and support productive HIV infection. For the HIV latency study, the HC69.5 line is a good model to examine mechanisms underlying the establishment and reactivation of HIV latency. For studying HIV interaction with other brain cells, microglia containing cerebral organoids (MCOs) are an appropriate model to determine how HIV infected microglia contribute to neuroinflammation, neuronal damage, and the development of HAND. In general, by leveraging the complementary strengths of different microglia models, researchers can gain a more comprehensive understanding of the complex interplays between HIV, microglia, and other brain cells. However, it is necessary to further improve and develop human microglia models for NeuroHIV studies. We have included the information above in the last paragraph of the Discussion section (Lines 241-249).
Comment 3: Authors should also comment on this explaining how the available models could be exploited to improve our understanding on the HIV -microglia (and CNS) interaction.
Response: We agree and have now added new comments in the manuscript (Lines 241-249).
Reviewer 2 Report
Comments and Suggestions for Authors
In this brief review, Sarkar et al. summarized the currently used human microglial models for NeuroHIV research. Overall, the review is concise and well written, listing major pros and cons of each microglial model used in NeuroHIV research, which is timely and potentially useful for researchers to select appropriate microglial model to address specific research questions.
A minor concern is that a brief introduction of the significance of NeuroHIV research may be needed for readers who are not familiar with NeuroHIV, which may strengthen the impact of the review.
Author Response
Reviewer: 2
We appreciate the reviewer’s positive comment,“Overall, the review is concise and well written, listing major pros and cons of each microglial model used in NeuroHIV research, which is timely and potentially useful for researchers to select appropriate microglial model to address specific research questions” .
Minor Concern: A minor concern is that a brief introduction of the significance of NeuroHIV research may be needed for readers who are not familiar with NeuroHIV.
Response: We agree and have added a brief introduction in the manuscript (Lines 33-40).